# Papillary Thyroid Microcarcinoma: Differences between Lesions in Incidental and Nonincidental Settings—Considerations on These Clinical Entities and Personal Experience

Giorgio Lucandri [1,*], Giulia Fiori [1], Francesco Falbo [1], Vito Pende [1], Massimo Farina [1], Paolo Mazzocchi [1], Assunta Santonati [2], Daniela Bosco [2], Antonio Spada [2] and Emanuele Santoro [1]

[1] Department of Surgical Oncology, San Giovanni-Addolorata Hospital, Via Dell'Amba Aradam 9, 00184 Rome, Italy; fiori.1466303@studenti.uniroma1.it (G.F.); ffalbo@hsangiovanni.roma.it (F.F.); vpende@hsangiovanni.roma.it (V.P.); mfarina@hsangiovanni.roma.it (M.F.); pmazzocchi@hsangiovanni.roma.it (P.M.); esantoro@hsangiovanni.roma.it (E.S.)

[2] Endocrinologic and Metabolic Departmental Ward Unit, San Giovanni-Addolorata Hospital, Via Dell'Amba Aradam 9, 00184 Rome, Italy; asantonati@hsangiovanni.roma.it (A.S.); dbosco@hsangiovanni.roma.it (D.B.); aspada@hsangiovanni.roma.it (A.S.)

[*] Correspondence: glucandri@hsangiovanni.roma.it; Tel.: +39-3398472308

**Abstract:** Papillary thyroid microcarcinoma (PTMC) represents 35–40% of all papillary cancers; it is defined as a nodule $\leq$ 10 mm at the time of histological diagnosis. The clinical significance of PTMC is still controversial, and it may be discovered in two settings: incidental PTMC (iPTMC), in which it is identified postoperatively upon histological examination of thyroid specimens following thyroid surgery for benign disease, and nonincidental PTMC (niPTMC), in which it is diagnosed before surgery. While iPTMC appears to be related to mild behavior and favorable clinical outcomes, niPTMC may exhibit markers of aggressiveness. We retrospectively review our experience, selecting 54 PTMCs: 28 classified as niPTMC (52%) and 26 classified as iPTMC (48%). Patients with niPTMC showed significant differences, such as younger age at diagnosis ($p < 0.001$); a lower male/female ratio ($p < 0.01$); a larger mean nodule diameter ($p < 0.001$); and a higher rate of aggressive pathological findings, such as multifocality, capsular invasion and/or lymphovascular invasion ($p = 0.035$). Other differences found in the niPTMC subgroup included a higher preoperative serum TSH level, higher hospital morbidity and a greater need for postoperative iodine ablation therapy ($p < 0.05$), while disease-free long-term survival did not differ between subgroups ($p = 0.331$) after a mean follow-up (FU) of 87 months, with one nodal recurrence among niPTMCs. The differences between iPTMC and niPTMC were consistent: patients operated on for total thyroidectomy and showing iPTMC can be considered healed after surgery, and follow-up should be designed to properly calibrate hormonal supplementation; conversely, niPTMC may sometimes exhibit aggressive behavior, and so the FU regimen should be closer and aimed at early detection of cancer recurrence.

**Keywords:** thyroid; microcarcinoma; incidental; nonincidental; cancer; ultrasound; fine needle; active surveillance; multifocality; thyroidectomy

## 1. Introduction

The incidence of well-differentiated thyroid carcinoma, particularly papillary thyroid cancer (PTC), has been increasing in industrialized countries for the last 20–30 years; epidemiologic studies and large surveillance database reports show more than a twofold increase in thyroid cancer since 1995 [1]. It certainly represents the most common cancer among endocrine tumors, but its overall incidence remains low, with a rate of 1.2–1.5% of all malignancies [2].

This increased incidence of PTC is likely due to increased diagnosis, attributable to the common use of tools such as neck ultrasound (US) and fine-needle aspiration

cytology (FNAC); identification of thyroid nodules may also be related to routine imaging performed for other pathologies, such as neck injuries, carotid diseases, and cervical diseases [3]. Furthermore, improvements in histologic techniques, the introduction of routine immunohistochemical staining and increased diligence in pathological analysis have contributed to this trend [4].

PTC is usually related to a favorable prognosis, with very high long-term disease-free survival and a 10-year survival rate of 93% [5]; thus, if PTC does not represent a public health problem because of its low incidence, favorable prognosis and low mortality, the increasing rate of new diagnoses cannot be underestimated.

Among newly diagnosed PTCs, lesions smaller than 10 mm in maximum diameter, known as papillary thyroid microcarcinoma (PTMC), represent 35–40% [6]. The term PTMC was firstly introduced by the World Health Organization (WHO) in 1989 [7]; it is defined as a lesion ≤ 10 mm in its largest diameter and hence classified as pT1a in the TNM Classification of Malignant Tumors, 8th Ed. [8].

The clinical significance of PTMC and its effects on patients' clinical outcomes and overall survival is still a matter of debate, as many authors consider PTMC a nonprogressive disease without any effect on survival rather than a real oncological entity. This has led many authors to propose protocols of active surveillance (AS) as an alternative to surgical removal in subsets of patients with favorable prognostic factors and good compliance to treat a comorbid suspicious nodule (low-risk PTMC) [9]. Patients may be promptly shifted to surgical treatment if alarming factors such as nodule enlargement, changes in the ultrasonographic pattern or aspects of multifocality appear during close follow-up [10–13]. PTMC typically shows this indolent behavior, but on the other hand, a small subset of PTMC (5–10%) may exhibit an aggressive attitude, with a trend toward capsular invasion, multifocality, nodal involvement and sometimes lymph node metastasis representing the first clinical sign of PTMC (occult PTMC) [14,15]. Confirmation of expected cancer, together with risk factors such as multifocality, capsular and vascular invasion, histological variants and mutated BRAF, may contribute to the selection of this subset of patients who are good candidates for closer follow-up and more radical surgery [16].

PTMC may be discovered in two main settings: incidental PTMC (iPTMC), in which it is identified postoperatively upon histological examination of thyroid specimens following thyroid surgery for benign disease [3,17], and nonincidental PTMC (niPTMC), in which it is diagnosed before surgery upon FNAC of small thyroid nodules detected on neck US or due to the presence of nodal metastasis [2,4,18,19]. iPTMC is usually related to a favorable outcome, with a low risk of recurrence; in contrast, niPTMC may exhibit more aggressive behavior, with a certain rate of LN metastases, higher rates of multifocality and bilaterality, locoregional recurrence during follow-up [6,19]. Diagnosis of iPTMC is usually a consequence of demolitive thyroid surgery for a diffuse thyroid pathology (i.e., total thyroidectomy for goiter or nodular thyroiditis), while niPTMC may also be diagnosed after thyroid-sparing surgery (i.e., lobectomy) performed for suspicious nodules detected by preoperative ultrasound and/or cytology. It could be reasonable to expect more aggressive features in the latter situation, in which the presence of the nodule itself usually represents the surgical indication [20].

To verify any difference between subgroups of patients exhibiting iPTMC and niPTMC, we retrospectively reviewed our experience in thyroid surgery for PTMC and compared results with large series reported in the recent literature; our last purpose was to investigate whether patients with incidental or nonincidental PTMC may be candidates for different therapeutic strategies and follow-up protocols.

## 2. Materials and Methods

From 1 June 2010 to 30 June 2022, 534 thyroid surgical procedures were performed at our Department of Surgical Oncology; out of this consecutive series, we first selected patients with a definitive histological finding of PTC (No 132, 24.71%), then those cases exhibiting a PTMC (No 54, 40.9% of PTCs; 10.11% of the entire series). A PTMC was defined

as a lesion $\leq 10$ mm in its largest diameter at the time of histological diagnosis and thus classified as pT1a Nx in the TNM Classification of Malignant Tumors, 8th Ed. [8]. In further evaluation, these patients were divided into two subgroups: those in whom PTMC was an incidental discovery after thyroid surgery performed for benign diseases (labeled iPTMC; 26 patients, 48.15%) and those in whom PTMC itself represented the indication for surgical removal according to preoperative FNAC and/or US nodular vascular patterns (labeled niPTMC; 28 patients, 51.85%).

Data concerning patients' demographics, clinical presentation, surgical modalities and pathological variables were observed, and any difference between these subgroups is emphasized herein. Patients with follicular or medullary tumors, as well as those with the noninvasive follicular thyroid neoplasm (NIFTP) entity, were excluded from the analysis.

A telephone survey analysis was carried out to track the length and status of follow-up; patients were asked about their survival, the presence of signs of locoregional or distant oncological evolution and their early surgical outcomes (vocal-cord dysfunction, signs of hypoparathyroidism).

### 2.1. Preoperative Assessment and Diagnostic Work Up

Patients affected by thyroid disease with any indication for surgical treatment are usually referred by the Endocrinologic and Diabetological Departmental Outpatient Service to our Department of Surgical Oncology; these patients are assigned to surgery after a weekly multidisciplinary meeting, in which the most appropriate treatment for each patient is discussed. Preoperative diagnostic workup always includes anesthesiologist ASA score assessment, a thyroid hormonal profile and B-mode/Doppler ultrasonography; any thyroid nodule with suspected US features and/or a peri- or intra-lesional vascular pattern is recommended for FNAC. Additional workup may include dosages of thyroid autoantibodies, calcitonin, thyroglobulin and PTH in suspicious nodules, as well as cervico-mediastinal CT scans in mediastinal goiter or whenever neck adenopathies are clinically or radiologically detectable. All patients undergo preoperative evaluation of vocal cords by an ENT specialist.

### 2.2. Surgical Treatment

All surgical procedures were performed by the same surgical team. Total thyroidectomy (TT), thyroid lobectomy (TL) and completion thyroidectomy (CT) were considered among the options; other incomplete procedures such as subtotal thyroidectomy, isthmectomy or simple enucleoresection were never performed.

At our institution, the site of the thyroid is always reached via an open route. A Harmonic Focus® (HF) shears device (Ethicon Endo-Surgery, Guaynabo, PR, USA) is routinely used for tissue dissection in partial procedures, closure of smaller blood vessels, grasping of tissue and cutting maneuvers, while terminal branches of superior thyroid artery are interrupted by using titanium hemostatic clips or Ethicon Vicryl™ 3/0 sutures. The recurrent laryngeal nerve (RLN) is typically identified behind the branches of the division of the inferior thyroid artery and smoothly dissected up to its point of entry into the laryngeal complex. The RLN is continuously identified by using intermittent intra-operative nerve monitoring (NIM Vital® Medtronic Xomed, Inc., Jacksonville, FL, USA), and an electromyographic recording is printed and attached, together with the operation report, to the patient's chart; both pre-dissection and post-dissection signals are recorded on each side. At least one parathyroid gland on each side is identified and preserved. Once the specimen is removed, hemostasis is verified, and Surgiflo™ Hemostatic Matrix (Ethicon—J&J MedTech) is always positioned on the surgical bed. Single tubular section drainage is placed into the site of the thyroid, and the integrity of the strap muscles and platysma is restored by interrupted absorbable sutures.

*2.3. Postoperative Care and Follow-Up*

Surgical drainage is usually left in place for 24 h and then removed; the patient is discharged on the second postoperative day. During the study period, serum calcium and phosphorus levels were measured on both the first and second postoperative days; oral calcium and vitamin D3 supplements were considered only in case of signs and symptoms of hypoparathyroidism. Whenever the patient presented hoarseness, any degree of vocal alteration or any form of tirage, a video laryngoscopy was performed by an ENT specialist and compared with the preoperative examination.

Patients are usually sent to our outpatient service for surgical evaluation on the seventh postoperative days. All thyroid specimens in this study were examined by the same two pathologists, and their weight, shape, color and cut surface were always described. The maximum diameter of each PTMC was examined and reported in mm, and the larger nodule was considered in the case of multicentric lesions. Formalin-fixed, paraffin-embedded tissue obtained from samplings was stained was hematoxylin and eosin; for immunohistochemical (IHC) analysis, sections were immunostained with CD34 and CD31 monoclonal antibodies. The histological diagnosis was made according to the WHO guidelines and AJCC 8th Ed. TNM Staging System [8]. The following pathological features were investigated: capsular invasion, presence of multifocal disease, lymphovascular invasion (LVI) and nodule location inside the thyroid gland. Each patient was re-referred to the endocrinological outpatient service for hormonal supplementation with levothyroxine, a follow-up protocol and/or evaluation by a nuclear medicine specialist.

*2.4. Statistical Analysis*

All patients' data were entered into an Office Excel worksheet (Microsoft Office 2021©) and analyzed by using the JASP© statistical software (0.17.1 Inc., Amsterdam Un. The Netherlands); descriptive analysis is expressed in terms of frequency, mean ± SD, median and range. All variables were matched to each other in a univariate analysis. The chi-square test and Fisher's exact test were used to compare variables; statistical significance was defined as $p < 0.05$. The Kaplan–Meier curve was used to estimate disease-free survival.

## 3. Results

The subgroups of patients carrying iPTMC and niPTMC were similar in size, with 26 (48.14%) and 28 (51.86%) patients, respectively, and appeared to be homogeneously distributed during the study period (Figure 1).

Their demographic characteristics are shown in Table 1.

**Table 1.** Demographic characteristics of patients.

|  |  | VALID | iPTMC | niPTMC | *p* |
|---|---|---|---|---|---|
| PATIENTS |  | 54 | 26 | 28 | - |
| AGE |  |  |  |  |  |
|  | Mean | 48.685 | 54.346 | 43.429 | <0.001 |
|  | SD | 11.922 | 12.570 | 8.557 |  |
|  | Range | 26–72 | 26–65 | 29–72 |  |
| GENDER |  |  |  |  |  |
|  | Male | 12 | 7 | 5 |  |
|  | Female | 42 | 19 | 23 |  |
|  | M/F ratio | 28.57 | 36.84 | 21.73 | <0.01 |

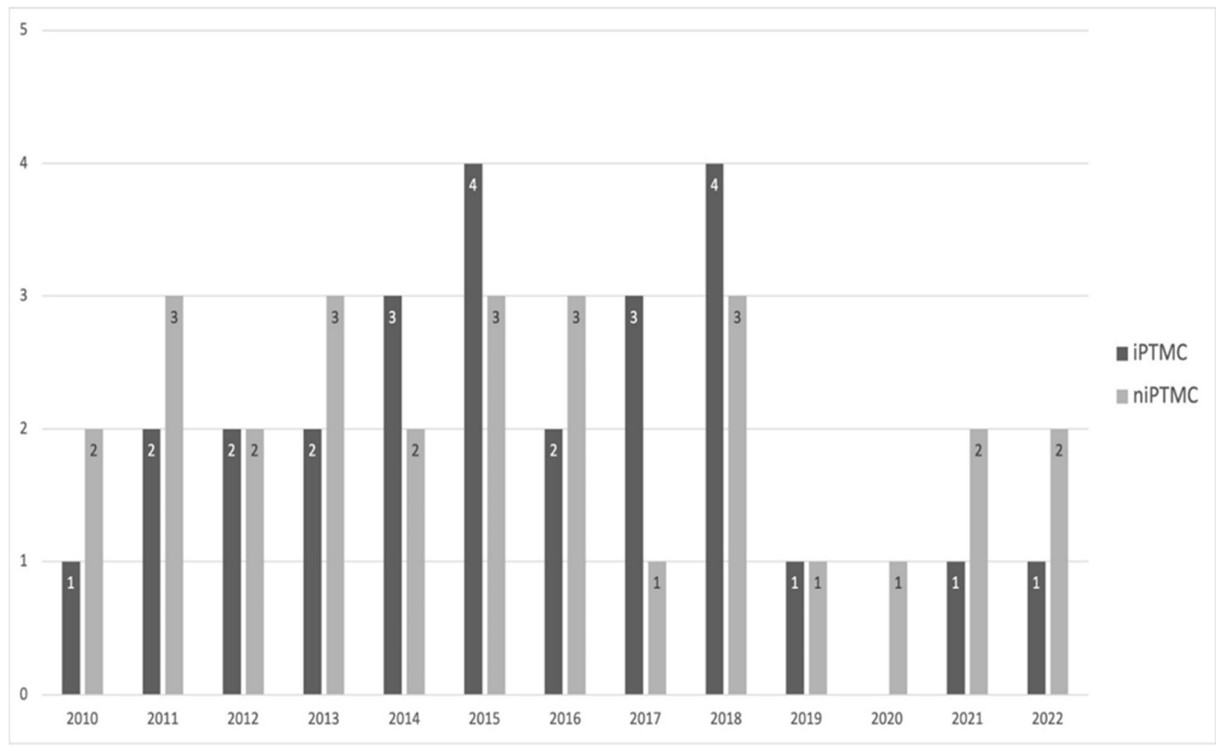

**Figure 1.** Distribution of patients operated on for PTMC during study period.

Significant differences were observed in the iPTMC subgroup, including older age and a higher M/F ratio. All patients received a preoperative routine cervical Doppler ultrasonography, while the iPTMC subgroup contained a significantly higher proportion of patients subjected to cervico-mediastinal CT scans than the other subgroup (9/26–34.61% vs. 3/28–10.71%, $p$ = 0.035); the need for preoperative assessment of cervico-mediastinal multinodular goiter may explain this significant difference. Preoperative median TSH levels were higher in the niPTMC subgroup, although the difference did not reach statistical significance (0.9 mIU/L, range 0.5–1.3, vs. 0.7 mIU/L, range 0.5–1, respectively; $p$ = 0.076). Concerning indications for thyroid removal (Table 2), a high prevalence of multinodular goiter was observed in the iPTMC subgroup (73.08%), while, in the niPTMC subgroup, a significant rate of patients carrying highly suspicious (TIR 4) or diagnosed (TIR 5) papillary carcinoma (53.57%) was reported; one patient with a TIR 3a nodule asked for surgical operation instead of an active surveillance (AS) protocol [21].

**Table 2.** Surgical indications according to preoperative diagnosis.

|  |  | **No** | **%** |
|---|---|---|---|
| iPTMC (n° 26) | Multinodular goiter | 19 | 73.08 |
|  | Thyroid hyperplasia | 3 | 11.54 |
|  | Chronic thyroiditis | 4 | 15.38 |
| niPTMC (n° 28) | TIR 3 (before 2015) | 5 | 17.86 |
|  | TIR 3a | 1 | 3.57 |
|  | TIR 3b | 7 | 25.00 |
|  | TIR 4 | 13 | 46.43 |
|  | TIR 5 | 2 | 7.14 |

All patients in the iPTMC subgroup received TT according to the first endocrinological indication. Patients with suspicious thyroid nodules were submitted to TT in twenty-two cases (78.57%), whenever the presence of contralateral nodules or tissue inhomogeneity could be demonstrated at preoperative diagnostic workup; the remaining six patients received TL (21.43%). CT was required for three of the latter patients (50%):

one patient presented both multifocal disease and capsular invasion at the time of definitive histopathological diagnosis, one patient had multifocal disease, while the remaining patient asked for completion surgery instead of active follow-up; reoperation was performed at 2, 4 and 5 months since first surgical approach. Thus, a total of 31 surgical procedures were performed on niPTMC subgroup. All patients in the niPTMC subgroup received central-compartment exploration, but only 15 patients with suspicious or diagnosed PTC (53.57%) underwent central-compartment neck dissection.

Three patients developed postoperative complications (5.55%): hoarseness and transient moderate vocal alterations in two cases and transient hypocalcemia in one case. These three patients all belonged to the niPTMC subgroup, received medical therapies, and made a complete recovery. All patients were discharged on the second postoperative day (POD).

At the time of definitive pathological examination, all PTMCs were staged as pT1a; concerning patients who underwent central-compartment neck dissection, a mean of 4.27 ± 1.25 nodes were resected (range 2–7); only one patient presented as N+ on histological examination (2/6) with definitive pT1a pN1a staging. Pathological findings (Table 3) demonstrated significant differences between subgroups, with niPTMC having a larger mean diameter ($p < 0.001$), a higher rate of multifocality and a higher rate of apparent capsular or lymphovascular invasion, considered together as aggressive findings ($p = 0.035$). Multiple markers were detected in 3 patients (5.55%). Most PTMCs from both subgroups showed an intraparenchymal location, while a certain proportion of niPTMCs arose from the peritracheal area or in proximity to the RLN, but the difference in distribution between subgroups was not significant.

**Table 3.** Tumor characteristics on histological examination and IHC analysis.

| | VALID (54) | iPTMC (26) | niPTMC (28) | *p* |
|---|---|---|---|---|
| Average size (mm) | | | | |
| Mean | 5.593 | 3.808 | 7.250 | <0.001 |
| SD | 2.574 | 2.227 | 1.578 | |
| Range | 1–10 | 1–9 | 3–10 | |
| Median | 7 | 3 | 8 | |
| Multifocality | | | | |
| No | 14 | 4 | 10 | |
| % | 25.926 | 15.385 | 35.714 | 0.089 |
| Capsular invasion | | | | |
| No | 2 | - | 2 | |
| % | 3.704 | - | 7.143 | 0.165 |
| LVI | | | | |
| No | 5 | 1 | 4 | |
| % | 9.25 | 3.84 | 14.28 | 0.186 |
| Markers of aggressiveness | | | | |
| No | 18 | 4 | 14 | |
| % | 33.33 | 15.38 | 50 | 0.035 |
| Tumoral location | | | | |
| Subcapsular | 6 (11.12%) | 2 (33.33%) | 4 (66.66%) | |
| Peritracheal | 4 (7.4%) | 1 (25%) | 3 (75%) | |
| Proximity to RLN | 5 (9.25%) | 2 (40%) | 3 (60%) | |
| Intraparenchymal | 39 (72.23%) | 21 (53.9%) | 18 (46.1%) | 0.423 |

All patients were scheduled and followed up according to the treatment protocol; they all had an 18-month minimum follow-up in order to test whether histopathological high-risk features could affect disease-free survival and definitive clinical outcomes. The mean follow-up was 87.643 ± 41.496 months (range 18–154) for the niPTMC subgroup and 90.346 ± 48.155 months (range 22–162) for the iPTMC subgroup, without a statistical

difference (*p* = 0.811). After oncological and nuclear medicine re-evaluation, nine patients (16.66%) exhibiting larger lesions, the presence of associated high-risk factors or nodal involvement were sent for diagnostic iodine total-body scanning, of whom seven patients belonged to the niPTMC subgroup and two belonged to the iPTMC subgroup (*p* < 0.05); four patients (all from the niPTMC subgroup) showing some form of cervical iodine uptake were subjected to $I^{131}$-iodine ablation therapy, with radioiodine activity ranging between 30 mCi (1.1 GBq) and 90 mCi (3.3 GBq). Disease-free survival did not differ significantly between subgroups (*p* = 0.331) and is presented as the Kaplan–Meier estimate in Figure 2.

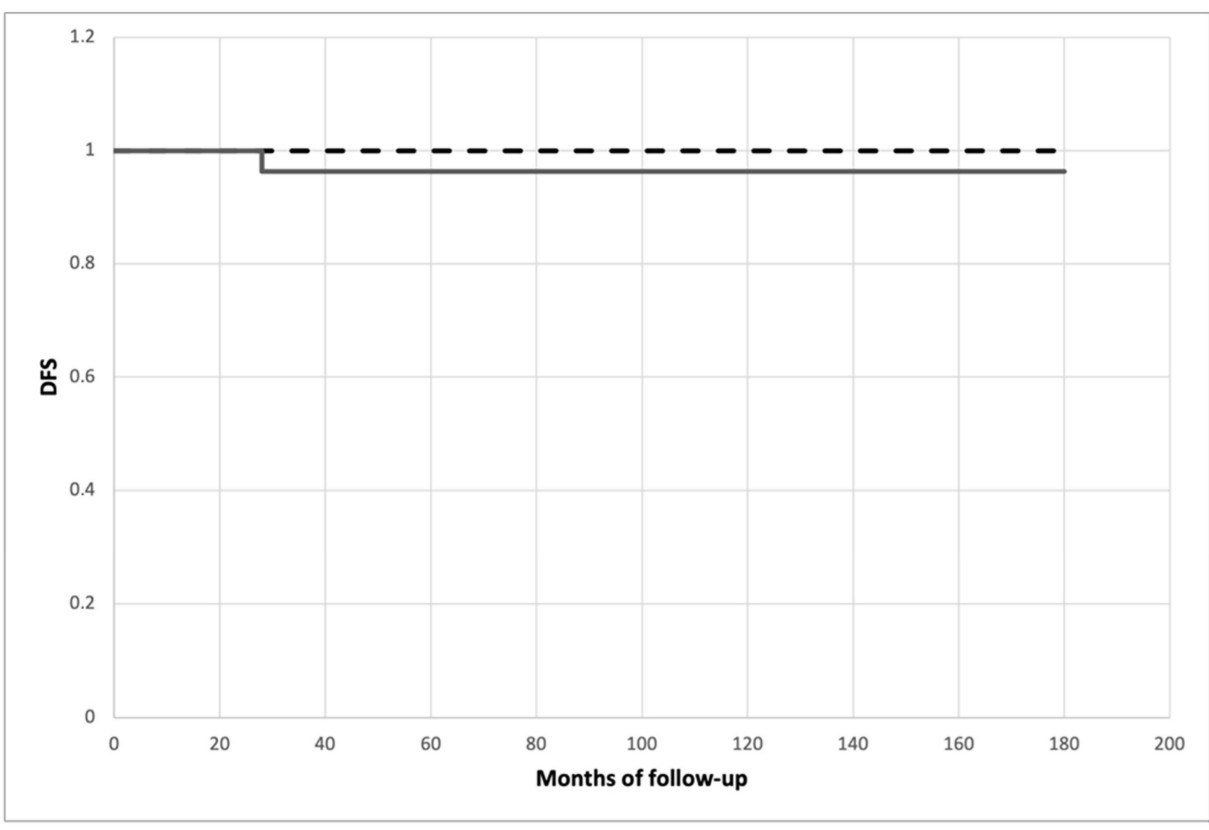

**Figure 2.** Kaplan–Meier plot of recurrence-free survival (*p* = 0.331). Dashed line: iPTMC. Solid line: niPTMC. DFS—Disease-free survival.

One patient from the niPTMC subgroup, previously subjected to TT and central node dissection for a TIR 5 nodule, developed clinical and radiological evidence of omolateral neck nodal metastases at 28 months after the surgical procedure and underwent conservative lateral neck lymphadenectomy. At present, this patient, similar to the others from both subgroups, does not show any sign of cancer recurrence. Two non-cancer-related deaths (one for each subgroup) were recorded at 142 and 153 months after the surgical procedure.

## 4. Discussion

During recent decades, several reports seemed to highlight more aggressive behavior in niPTMC than in incidental papillary micromalignancies; this aggressive attitude includes higher rates of capsular invasion, multifocality and bilaterality as well as a trend toward developing lymph node metastases [19,22–24]. Differences were found also in demographic information, such as younger age, a lower male/female ratio and higher preoperative serum TSH levels [18,19]. The appearance of papillary micromalignancies in these two different settings with such different clinical and pathological findings leads one to hypothesize the presence of two different diseases rather than different forms of the same tumor [6].

The first consideration regards the mean diameter of PTMC, which appears significantly higher for niPTMC in all reported series [4,18,19,22–24]; similar rates of iPTMC smaller and larger than 5 mm. have been reported by Kaliszewski et al., while rates of 15% and 85%, respectively, have been found for niPTMC ($p < 0.001$) [18]. It could appear reasonable to relate markers of aggressive behavior to an increased diameter of thyroid nodules; larger volume makes thyroid nodules more detectable on US, particularly if they develop within a homogeneous parenchyma. Moreover, increased diameter could be related to capsular involvement and to tracheal infiltration, above all for nodules located in proximity to the RLN or in the peritracheal area [10]. On the other hand, the presence of multinodular goiter makes US diagnosis of a suspicious nodule more difficult as the diameter decreases. In addition, our experience confirms this finding, with the mean diameter of niPTMC being double that of iPTMC (7.2 vs. 3.8 mm; $p < 0.001$).

Some pathogenetic role of increased THS levels in promoting cancer progression has been supposed; this finding could explain the significantly higher level in niPTMC reported by Provenzale et al. in their consecutive series (1.1 vs. 0.6 mIU/L, $p < 0.0001$) [19]; on the other hand, the development of thyroid autonomy observed in multinodular goiter appears to be related to decreased TSH level. Similarly, in our cohort, we found higher preoperative TSH levels in the niPTMC subgroup, although the values did not reach statistical significance ($p = 0.076$).

Concerning other pathological findings, we emphasize the experience of Provenzale et al., who retrospectively reviewed their consecutive series dealing with PTMC, selecting two subgroups consisting of iPTMC (No 92) and niPTMC (No 67). These subsets of patients significantly differed in mean age at diagnosis, with older patients for iPTMC than niPTMC (respectively, 53.3 ± 13.2 years vs. 44.9 ± 14.8 years, $p = 0.0002$); iPTMC patients also had a smaller tumor size (median 4 mm vs. 9 mm, $p < 0.0001$) and a higher frequency of multifocality (76.1% vs. 52.2%, $p = 0.001$). They failed to demonstrate a significant difference concerning the frequency of nodal metastases [19]. Contemporary experience from Poland, comparing 37 iPTMC (38.14%) and 60 niPTMC (61.86%), showed in the former subgroup a slower, although not significantly lower, rate of patients younger than 45 years ($p = 0.205$), while niPTMC exhibited a higher rate of cervical node involvement (18.33%, $p < 0.001$). No significant differences could be demonstrated for gender prevalence ($p = 0.422$) or multifocality ($p = 0.122$) [18]. In the largest cohort reported to date, 527 iPTMC (66%) and 134 niPTMC (17%) were reviewed, adding as a third subgroup 142 cases (18%) of niPTMC symptomatic of nodal and/or distant metastases; younger median age, larger tumor size and higher frequencies of extrathyroidal extension and nodal metastases were reported in niPTMC, both for nonincidental nodules and for metastatic disease ($p < 0.05$). No significant differences were demonstrated in crude or recurrence-free 5- or 10-year survival [4]. Data on surgical treatment of PTMC, such as rates of TT, TL and associated nodal dissection, are often not specified in these manuscripts; knowledge about the type of surgery might contribute to explaining the results on residual disease or the high rate of nodal metastases, above all in the niPTMC subgroup. For some findings, such as age at diagnosis and rate of multifocality, results are conflicting and do not uniformly support one conclusion [18,19].

We usually receive patients with suspicious nodules on US or FNAC from our Endocrinological Outpatient Service; patients with low follicular proliferating nodules (TIR 3A) are recommended for an AS protocol, while patients exhibiting high follicular proliferation (TIR 3B), as well as those with highly suspicious (TIR 4) or certain (TIR 5) PTC, are recommended for surgery [21]. The volume and diameter of the nodule do not play a key role in this phase of the decision-making process as long as a cytological diagnosis has been acquired; very low surgical morbidity rates reported in our entire experience with thyroid surgery (permanent vocal cord dysfunction and hypoparathyroidism in less than 2‰) support this behavior. Each patient's choice and compliance are always taken into account in this therapeutic pathway. According to current guidelines, AS involves clinical evaluation, neck ultrasound and blood samples every 6 months for the first 2 years and then yearly [9,11–13,25,26]. Nodules in proximity to the trachea or the recurrent laryngeal nerve,

those arising in subcapsular sites and those exhibiting high-grade findings on FNAC do not meet the inclusion criteria for AS; during the surveillance period, loss of nodule stability (enlargement, changes in ultrasonographic pattern) or the appearance of nodal enlargement or multifocality leads to a sudden shift of the patient into surgical management [3,12,27].

In our experience, PTMC represents 40% of all papillary thyroid carcinomas; these data reflects what has been reported in the literature, with incidental and nonincidental PTMCs subgroups showing similar rates (48% and 52%). Differences between incidental and nonincidental lesions reached high consistency and concerned demographical, clinical and histopathological findings; statistical significance was found for younger age ($p < 0.001$), a lower male/female ratio ($p < 0.01$), larger nodule diameter ($p < 0.001$), higher detection of aggressive findings at histology ($p = 0.035$) and a higher need for postoperative RAI ($p < 0.05$) in the niPTMC subgroup. Some other differences may be explained by different therapeutic approaches: the higher morbidity in the niPTMC subgroup (10.71%) results from the rate of associated central node dissection (53%) and CT (11.5%). No reoperations were required on patients with postoperative complications; they were treated with medical therapy and showed prompt recovery. Additionally, in our experience, the subset of patients exhibiting aggressive PTMC seems to fall almost entirely within the niPTMC subgroup [28,29]. The unusually low rate of nodal recurrence, observed in both subgroups (respectively, 0% and 3.57% for iPTMC and niPTMC) probably represents the consequence of our aggressive attitude in treating TIR 4 and TIR 5 lesions with TT and prophylactic central neck dissection; the absence of thyroid-sparing surgery and adequate locoregional disease control resulted in excellent long-term disease-free survival, without consequences for patients' early outcomes.

## 5. Conclusions

According to the current literature and personal experience, incidental and nonincidental PTMC should be considered two different entities rather than different expressions of the same tumor.

Patients with niPTMC are usually younger and more frequently female; show higher preoperative serum TSH levels; have a larger nodule diameter at the examination of the specimen; and show a higher rate of malignant expression, such as capsular and/or lymphovascular invasion, multicentricity and bilaterality; these patients also exhibit a certain rate of nodal involvement [30]. As an additional consideration, patients with niPTMC are usually treated according to the results of preoperative FNAC and type of US vascular pattern: such data, together with age consideration, may lead to a thyroid-sparing surgery attitude, mainly on TIR 3b nodules (TL). As definitive histology is obtained, the presence of larger nodules, as well as signs of aggressive behavior, should direct physicians to a close follow-up regimen on these patients or to a CT. We usually subject patients carrying niPTMC to a close follow-up regimen, aimed at early discovery of possible residual disease or nodal recurrence.

On the other hand, patients with iPTMC are older and have just been subjected to TT, according to the presence of multinodular goiter or other diffuse thyroid pathologies; in these cases, iPTMC usually exhibits smaller size and the absence of an aggressive attitude, and so the type of surgical operation may be reasonably considered definitive and the patient healed. Such a picture should refer the patient with iPTMC to a simple follow-up protocol for periodic clinical checks and hormonal supplementation; in such patients, we usually aim for postoperative controls to optimize hormonal levels and not to search for possible recurrence.

Despite these different findings, whenever tailored follow-up protocols are properly applied, both iPTMC and niPTMC are related to a favorable clinical outcome; in fact, if the presence of two different settings may result in different therapeutic choices and follow-up regimens, any consequence would have significant implications for patients' quality of life and overall long-term survival.

**Author Contributions:** Conceptualization, G.L. and G.F.; methodology G.L., V.P. and P.M.; software F.F.; validation M.F., A.S. (Antonio Spada) and D.B.; formal analysis F.F.; investigation A.S. (Antonio Spada); resources M.F.; data curation F.F., writing-original draft preparation G.L., writing-review and editing G.L. and P.M.; visualization F.F.; supervision E.S. and A.S. (Assunta Santonati); project acquisition G.L.; project administration M.F. All authors have read and agreed to the published version of the manuscript.

**Funding:** This research received no external funding.

**Institutional Review Board Statement:** Ethical review and approval were waived for this study due to its retrospective nature.

**Informed Consent Statement:** Informed consent was obtained from all subjects involved in the study.

**Data Availability Statement:** The data presented in this study are available on request from the corresponding author.

**Conflicts of Interest:** The authors declare no conflict of interest.

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
