# Peer review of "Papillary Thyroid Microcarcinoma: Differences between Lesions in Incidental and Nonincidental Settings—Considerations on These Clinical Entities and Personal Experience"

_curroncol, doi:10.3390/curroncol31020070_

Round 1

Reviewer 1 Report (Previous Reviewer 2)

Comments and Suggestions for Authors

The revised version of the manuscript is considerably improved, I have no further issues to raise at his point

Reviewer 2 Report (Previous Reviewer 3)

Comments and Suggestions for Authors

Comments on the Quality of English Language

Reviewer 3 Report (New Reviewer)

Comments and Suggestions for Authors

Dear Authors,

I find  your manuscript very interesting.

The presence of two different settings may result in different therapeutic choices and follow up regimens, any consequence significantly affect patient’s quality of life and overall long term survival. 

This manuscript is a resubmission of an earlier submission. The following is a list of the peer review reports and author responses from that submission.

Round 1

Reviewer 1 Report

Comments and Suggestions for Authors

Papillary thyroid cancer is common and the number and proportion of small (including microcarcinomas) and low-risk tumors is increasing. Therefore, the proper treatment approach in terms of aggressiveness is important. The authors tackle the issue of the incidental PTMCs found in surgical specimens after thyroidectomy for benign conditions. The main still unanswered question is what to do with these patients. The topic of the manuscript is interesting, though benign goitres are being removed less often in the recent years.

There are some issues that need to be addressed:

1.     The sample size is small and reliable analysis with convincing conclusions are difficult to achieve.

2.     Some particular comments and questions:

Materials and methods:

Lines 102-109: The authors' definition of PTMC included absence of capsular involvement on US. Further on, the PTMC were divided into niPTMC and iPTMCs. iPTMCs were incidentally found in the surgical specimens. Therefore, they would not have been described on pre-surgical US. How would they fulfil the criteria for PTMC?

Lines 116-118: Do the authors rely on a phone interview? Locoregional assessment requires at least US evaluation.

Why would lesions with peri- or intranodular vascular pattern be considered for FNA? The majority of nodules would exhibit either peri- or intramodular pattern.

Thyroid lodge or thyroid bed?

Statistical processing

How were continuous variables analyzed? Please indicate. I would expect a non-parametric test, as the groups are small and some of the data indicate non-normal distribution.  

Results

Line 207 Please check the significance in the gender distribution as p might not be below 0.05.

The definition of the PTMC in the Introduction excludes lesions with capsular and lymph node involvement. Both LN and capsular involvement were found in several of the subjects. Shouldn’t they be excluded from the PTMC group and all further analyses?  

The difference between TSH levels was not significant, e.i. the authors cannot conclude that TSH differed between the two groups.

Lines 233-244. The paragraph is not written very clearly. What does the CT have to do with the surgical decision? How many central lymph node explorations/dissections were done after all? Please, rewrite.

Table III: The nodule size distribution might deviate from normality. Please, check and present as medians if necessary.

The study period was defined to be June 2010 to June 2022 - 12 years. The range of the follow up (lines 281-284) reaches 169 and 172 months and that is more than 14 years. Please, correct the discrepancy or clarify. By the way is that an IQ range or full range?

What were the high-risk factors that contributed to the decision to do RAI ablation? A table with the histological variants would be interesting to the reader.

Discussion

It should be mentioned that the major difference between the two groups was that in the iPTMC no cancer was suspected and in the niPTMC cancer was expected. A better descriptive presentation of the iPTMCs might give a better idea. A microcarcinoma that is 1-2 mm would almost certainly be missed on most US exams of a multinodular goitre or some Hashimoto glands.        

Line 341: Why would the higher rate of cervical node involvement in niPTMC be surprising?

Lines 360-361. Please, remove the per thousand data. The cohort is small and having no permanent RLN or PT damage means just that.

line 370: Please, support the statement.

Lines 384-387: The authors refer to their experience but cite other research groups. Please support or rephrase.

The conclusions: the authors did not find difference in the recurrence rate in the two groups for the particular duration of the follow up. What is the reason for advising a different long-term approach?

Moreover, the therapeutic approaches in the two groups were quite similar - thyroidectomy, though the reasons were different.

In what risk group would the authors classify their patients? Current DTC treatment approaches are based on risk stratification. Do most of the subjects in both groups fall in the same (low-risk) category or not?

The tables are in a disarray. Please, straighten them.

There is no such word as significative. 

Comments on the Quality of English Language

The text should undergo English language proof reading.

Author Response

REPLIES TO COMMENTS OF REVIEWER n°1

We thank You for Your very interesting and remarkable comments, which helped us to improve and revise our manuscript; herein the replies to Your comments:

Our study design was extremely simple and easy to understand: to demonstrate that incidental and nonincidental PTMC radically differed for demographic, clinical and histopatological features, so depicting two different entities, rather than different aspects of the same disease. Our results clearly support this statement: the two cohort significantly differed for mean age at diagnosis, male/female ratio, mean diameter of the lesion, presence of histological markers of aggressiveness, need for postoperative RAI. Differences have been found for other features, such us preoperative TSH level and  singolar histological parameters, even if in these cases they did not reach statistical significance; lack of significance for some features is not hiden and represents a result itself: other experiences reported on current literature, with a similar design and indicated in our bibliography, came to similar conclusions, even if significativity was not reached for all parameters. Differences were also found for aspects such as need for completion thyroidectomy, postoperative morbidity, nodal involvement at histology and recurrence rate at follow up.  Our study aimed to demonstrate that these differences justify different follup up protocol: patients carring iPTMC are healed from cancer after surgical treatment of thyroid benign disease and they have to be followed only for hormonal supplementation; conversely patients submitted to thyroid surgery for suspicious or diagnosed PTC simply need for a closer and oncological follow up, because of an higher rate of aggressive markers at histology, a certain rate of nodal involvement, an higher need for iodine therapy and a certain rate of regional recurrence. In spite of these consistent differences, if both subgroups are properly followed up, their long term outcome is favorable, without significative differences between subgroup. Our sample size (54 patients with PTMC, 26 with iPTMC and 28 with niPTMC) results similar to that reported on other experiences and allowed us to test significative differences for several features.

MATERIALS AND METHOS:

Among iPTMC subgroup,  diagnosis is made on thyroid specimen. In these cases only histological features are taken into account. It is a matter of fact. Our main criteria for defining PTMC is size, as original definition by 1989 WHO.

As clearly written in the text, after surgical treatment our patients are addressed to our Endocrinological outpatient Service for follow up; they undergo blood analyses and neck US according to their follow up schedule. Whenever indicated, patients are re-addressed to our surgical unit for completion thyroidectomy or nodal dissection. Our phone survey was only aimed to update patients status and allowed us to detect 2 thyroid cancer unrelated deaths.  

As written in the text, US features other than vascular pattern (calcification, irregular border, hypoechogenicity, taller-than-wide shape) address patient to FNAC.

Thyroid lodge refers to an anatomical area reached via cervical route for ablation; thyroid bed refers to the surgical area which remains after thyroidectomy.

STATISTICAL PROCESSING

Parametric test have been used to analyze variables and to verify data distribution.

RESULTS

Concerning gender distribution, we emphasized statistical significance between male/female ratio, with higher value for iPTMC subgroup.

According to UICC/AJCC 8th ed., presence of thyroid pericapsular infiltration without crossing the capsule and involving  extrathyroid structures, such strap or scm muscles or cervical vessels, does not result in an upstaging of the PTMC. Pericapsular infiltration is considered among markers of aggressiveness of PTMC (like multifocality or lymphovascular invasion) in current Literature. Only evidence of capsular infiltration with involvementat of surrounding structures at preoperative US should be considered as exclusion criteria for PTMC definition.

Concerning preoperative TSH level, we did not come to any conclusion, but simply reported mean hormonal level in both subgroups.

CT has been performed whenever markers of aggressiveness, such pericapsular invasion or multicentricity contraindicate simple follow up of the remaining thyroid tissue. One patients, after histological diagnosis, claimed for CT because of psychological instability.

Medians as been added on Table III

We apologize for discrepancy concerning lenght of follow up; it has been recomputed and corrected in the text. Please consider that study period ended on June 2022 and minimum follow up is 18 months. This interval has to be added to the study period; at present maximum follow up is 154 months for niPTMC subgroup and 162 for iPTMC subgroup.  The range of follow up is expressed as full range

Aim of our study was not to analyze different therapeutical options for treating Papillary Thyroid Carcinoma (radiofrequency ablation, targeted therapies, radioiodine ablation therapy). As written in the text, patients are assigned to surgery after a weekly multidisciplinary meeting; RAI has been applied after surgical treatment, whenever total-body scanning showed some form of cervical iodine uptake.

DISCUSSION

Demographics, clinical and histopathological characteristics of iPTMC has been extensively described on text and table; moreover editor asked us for a significative reduction of lenght of the manuscript.

The term surprising has been removed from the text.

The very small rate of morbidity is clearly referred to our entire experience which deals with more than 700 thyroidectomy up to date; this rate justify our behaviour, so it has been left in the text but better specified.

As you righly say, the present approach to DTC largely depends on risk stratification, but this was not the matter of the present study; the so-called low-risk patients (low follicular proliferation, no nodule enlargement at US, absence of aggressive citology, absence of multifocality, low Ki67) are usually addressed to active surveillance protocol and so do not belong to our niPTMC cohort.

During uploading of our manuscript (written in a World template as requested by Curr. Onc.) tabular blocks of the Tables have been lost. We apologize for the inconvenience, which should have been corrected by the Editor before addressing the manuscript to the reviewers. Tables will be upload in correct form in the revised manuscript.

Manuscript has been firstly corrected by using the grammar checker app Grammarly, then revised by a native English speaker.

We are sure You may appreciate our efforts in answering to Your comments, in improving the manuscript and hope in a favorable judgment.

Best Regards

Giorgio Lucandri

MD PhD

AOSGA – ROME, ITALY

Reviewer 2 Report

Comments and Suggestions for Authors

This is a retrospective single-center study on a small sample of operated patients aiming to assess the differences between incidental and non-incidental papillary thyroid microcarcinoma PTMC.

The study design and results presentation are interesting and deserve to be presented but need to be improved.

The introduction needs to be massively shortened to the most relevant information, the rest can be part of the discussion.

Tables need to be reshaped to be easier to follow (as they are the columns are not aligned)

Reasons for surgery in iPTMC group needs further detail (esp thyroid hyperplasia? and chronic thyroiditis)

Study limitations (e.g. small study sample) need to be discussed.

Author Response

REPLIES TO COMMENTS OF REVIEWER n°2

We thank You for Your very interesting and remarkable comments, which helped us to improve and revise our manuscript; herein the replies to Your comments:

Our study design was extremely simple and easy to understand: to demonstrate that incidental and nonincidental PTMC radically differed for demographic, clinical and histopatological features, so depicting two different entities, rather than different aspects of the same disease. Our results clearly support this statement: the two cohort significantly differed for mean age at diagnosis, male/female ratio, mean diameter of the lesion, presence of histological markers of aggressiveness, need for postoperative RAI. Differences have been found for other features, such us preoperative TSH level and  singolar histological parameters, even if in these cases they did not reach statistical significance; lack of significance for some features is not hiden and represents a result itself: other experiences reported on current literature, with a similar design and indicated in our bibliography, came to similar conclusions, even if significativity was not reached for all parameters. Differences were also found for aspects such as need for completion thyroidectomy, postoperative morbidity, nodal involvement at histology and recurrence rate at follow up.  Our study aimed to demonstrate that these differences justify different follup up protocol: patients carring iPTMC are healed from cancer after surgical treatment of thyroid benign disease and they have to be followed only for hormonal supplementation; conversely patients submitted to thyroid surgery for suspicious or diagnosed PTC simply need for a closer and oncological follow up, because of an higher rate of aggressive markers at histology, a certain rate of nodal involvement, an higher need for iodine therapy and a certain rate of regional recurrence. In spite of these consistent differences, if both subgroups are properly followed up, their long term outcome is favorable, without significative differences between subgroup. Our sample size (54 patients with PTMC, 26 with iPTMC and 28 with niPTMC) results similar to that reported on other experiences and allowed us to test significative differences for several features.

Lenght of manuscript has been shortened to less than 4,000 words, by reducing both introduction and methodology.

During uploading of our manuscript (written in a World template as requested by Curr. Onc.) tabular blocks of the Tables have been lost. We apologize for the inconvenience, which should have been corrected by the Editor before addressing the manuscript to the reviewers. Tables will be upload and reshaped in correct form in the revised manuscript.

Indications for thyroidectomy on iPTMC subgroup has been depicted on Table II. Patients are usually addressed from Endocrinological ward unit to Surgery whenever thyroid enlargement cannot be conservatively managed, for appearance of symptoms or radiological signs such tracheal deviation.

Our cohort of PTMC was similar to that reported on other experiences in current Literature and allowed us to reach statistical significance for several features, such age, male/female ratio, size of PTMC, presence of markers of aggressiveness, need for postoperative RAI. Moreover patients were equally distributed into subgroups and lenght of follow up led us to state definitive conclusions.

As shown on Table III, a significative higher rate of patients belonging to niPTMC subgroup exhibited histopathological markers of aggressiveness (pericapsular invasion, LVI, multifocality), when compared to iPTMC cohort  [respectively 14/28 (50%) vs 4/26 (15.38%) p = 0.0352]. Multiple markers have been detected on 3 patients (5.55%).

We are sure You may appreciate our efforts in improving the manuscript and hope in a favorable judgment.

Best Regards

Giorgio Lucandri

MD PhD

Reviewer 3 Report

Comments and Suggestions for Authors

Dear authors,

I have reviewed your manuscript entitled ‘Papillary thyroid microcarcinoma: different settings between incidental and non incidental lesions – Considerations on these clinical entities and personal experience.

The aim of this retrospective study was to determine if there were significant differences in terms of clinical behavior and pathologic patterns between two groups of microcarcinomas: incidental ones found on pathologic examination and non incidental, diagnosed or suspected preoperatively and for which surgery was indicated.

The authors maintain that the two entities are of different nature and that the follow-up regimen should thus be adapted.

Main qualities of the work: long follow-up period and evidence that age at diagnosis and male/female ratio are different between the two groups, supporting the theory that there are truly two entities.

Main defaults:
-      Size of the sample studied: too small, probably explaining the absence of multivariable analysis

-       Choice of the 2017 WHO classification, obsolete

-       Manuscript is much too long to come up finally with some evidence only on age at diagnosis and male/female ratio

-       Irrationality of the reasoning: the authors frequently state that differences are not statistically different between the two groups but carry on with considering that these differences are meaningful

Introduction section

Page 2, lines 59 to 73: don’t forget to mention thermal ablation as a possibility in the management of PTMCs. This is included in ETA guidelines and international consensuses.
Page 2, line 84: “demolitive”: meaning ?
Page 2, lines 85-90: please explain in what circumstances a subcentimeter nodule should be FNAed.

Materials and Methods section

Page 3, lines 103 to 109: PTMC is defined not only by its size but also by the absence of capsular involvement and cervical lymphadenopathy. Please state that this definition is the one by the authors and not the 2017 WHO classification which only relies on size. Thus, one big problem with the manuscript appears: why did the authors elect to exclude patients with more aggressive PTMCs (lymph nodes, capsular involvement..) ?

Page 3, paragraph 2.1: remove, lacking interest

Pages 3 and 4: paragraph 2.2: remove most of it, no interest
Page 3: lodge:  ???

Results section

Page 5, Table 1: M/F ratio: 28.57 and all other figures: please check

Page 5: lines 211-213: this sentence (‘need for…significative difference’) belongs in the discussion section, not here.
Page 5, line 218: TIR 4, TIR 5: should have been mentioned and explained in the Methods section.

Page 6, lines 233-244: difficult to understand, seem there are contradictions in the figures. Please clarify.
Page 6, line 253: ‘clear differences’. Typical examples of contradiction with the figures shown in Table III where multifocality and capsular invasion were not significantly different between the two groups. Only size was and this is quite obvious because if the size was the same between the two groups, the PTMCs would all have been detected with US. Very small PTMC’s can’t be easily seen with US when millimetric or so.

Discussion section
Page 9, lines 354-369: remove because off topic. Same for conclusion section lines 402-409.

Comments on the Quality of English Language

English revision required

Author Response

REPLIES TO COMMENTS OF REVIEWER n°3

We thank You for Your very interesting and remarkable comments, which helped us to improve and revise our manuscript; herein the replies to Your comments:

Our study design was extremely simple and easy to understand: to demonstrate that incidental and nonincidental PTMC radically differed for demographic, clinical and histopatological features, so depicting two different entities, rather than different aspects of the same disease. Our results clearly support this statement: the two cohort significantly differed for mean age at diagnosis, male/female ratio, mean diameter of the lesion, presence of histological markers of aggressiveness, need for postoperative RAI. Differences have been found for other features, such us preoperative TSH level and  singolar histological parameters, even if in these cases they did not reach statistical significance; lack of significance for some features is not hiden and represents a result itself: other experiences reported on current literature, with a similar design and indicated in our bibliography, came to similar conclusions, even if significativity was not reached for all parameters. Differences were also found for aspects such as need for completion thyroidectomy, postoperative morbidity, nodal involvement at histology and recurrence rate at follow up.  Our study aimed to demonstrate that these differences justify different follup up protocol: patients carring iPTMC are healed from cancer after surgical treatment of thyroid benign disease and they have to be followed only for hormonal supplementation; conversely patients submitted to thyroid surgery for suspicious or diagnosed PTC simply need for a closer and oncological follow up, because of an higher rate of aggressive markers at histology, a certain rate of nodal involvement, an higher need for iodine therapy and a certain rate of regional recurrence. In spite of these consistent differences, if both subgroups are properly followed up, their long term outcome is favorable, without significative differences between subgroup. Our sample size (54 patients with PTMC, 26 with iPTMC and 28 with niPTMC) results similar to that reported on other experiences and allowed us to test significative differences for several features.

The 1989 WHO histological classification has been mentioned because firstly introduced the term Papillary Thyroid Microcarcinoma and defined the criteria that fulfilled this definition. This citation seems more than appropriate. All experiences reported on current Literature remind to WHO classification to properly define the PMTC entity.

Lenght of manuscript has been shortened to less than 4,000 words, as requested by Editorial Board.

Incidental and non incidental PTMC subgroups significantly differed for mean age at diagnosis, male/female ratio, mean size of ther lesion, presence of markers of aggressiveness (multifocality, capsular invasion, limphovascular invasion), need for postoperative RAI. These results seem to justify a different approach to these patients, when considering follow up protocols.

INTRODUCTION SECTION

 Aim of our study was not to analyze different therapeutical options for treating Papillary Thyroid Carcinoma (radiofrequency ablation, targeted therapies, radioiodine ablation therapy). As written in the text, patients are assigned to surgery after a weekly multidisciplinary meeting; in our experience patients not suitable for surgery because of multiple comorbidity or advanced aged, are addressed to radiologists for treating thyroid nodules by thermal ablation.

Patients carring diffuse thyroid pathologies are submitted to total thyroidectomy; the term demolitive refers to non-sparing surgery.

As written in the text, US features such as vascular pattern, calcification, irregular border, hypoechogenicity, taller-than-wide shape usually  address patient to FNAC; this decision-making procedure is unrelated to nodule diameter.

MATERIALS AND METHODS SECTION

Definition of PTMC mainly consider mean diameter of the lesion. Current literature often deals with subgroups of patients with PTMC, who exhibit capsular invasion (without crossing to surrounding structures) and/or nodal involvement (5-10%). Also in our serie patients with histopathological evidence for capsular invasion or nodal involvement are not excluded, as reported in the text and on Table III.

Thyroid lodge refers to an anatomical area reached via cervical route for ablation; thyroid bed refers to the surgical area which results after thyroidectomy.

RESULTS SECTION

Thyroid nodules are usually classified at cytology according to 2014 updated SIAPEC classification; we believe this is a data accepted worldwide and it does not need additional explanation. It is also cited on Bibliography (n°21).

The statement  “clear differences” is referred to our results between subgroups concerning mean age, male/female ratio, mean size of the lesion, presence of histological aggressive findings, need for recurrence to postoperative RAI. Also other experiences reported on Literature and cited on our Bibliography failed to demonstrate statistically significant differences for some histological or IHC findings between incidental and nonincidental PTMC.

Manuscript has been firstly corrected by using the grammar checker app Grammarly, then revised by a native English speaker.

We are sure You may appreciate our efforts in improving the manuscript and hope in a favorable judgment.

Best Regards

Giorgio Lucandri

MD PhD

AOSGA-ROME

ITALY